# Self-Constructed Deep Fuzzy Neural Network for Traffic Flow Prediction

**Jiyao An \***(ID)**, Jin Zhao, Qingqin Liu, Xinjiao Qian and Jiali Chen**

College of Computer Science and Electronic Engineering, Hunan University, Changsha 410012, China
\* Correspondence: jt_anbob@hnu.edu.cn

**Abstract:** Traffic flow prediction is a critical component of intelligent transportation systems, especially in the prevention of traffic congestion in urban areas. While significant efforts have been devoted to enhancing the accuracy of traffic prediction, the interpretability of traffic prediction also needs to be considered to enhance persuasiveness, particularly in the era of deep-learning-based traffic cognition. Although some studies have explored interpretable neural networks from the feature and result levels, model-level explanation, which explains the reasoning process of traffic prediction through transparent models, remains underexplored and requires more attention. In this paper, we propose a novel self-constructed deep fuzzy neural network, SCDFNN, for traffic flow prediction with model interpretability. By leveraging recent advances in neuro-symbolic computation for automatic rule learning, SCDFNN learns interpretable human traffic cognitive rules based on deep learning, incorporating two innovations: (1) a new fuzzy neural network hierarchical architecture constructed for spatial-temporal dependences in the traffic feature domain; (2) a modified Wang–Mendel method used to fuse regional differences in traffic data, resulting in adaptive fuzzy-rule weights without sacrificing interpretability. Comprehensive experiments on well-known traffic datasets demonstrate that the proposed approach is comparable to state-of-the-art deep models, and the SCDFNN's unique hierarchical architecture allows for transparency.

**Keywords:** intelligent transportation system (ITS); traffic flow prediction; hierarchical fuzzy inference systems; fuzzy neural network; modified Wang–Mendel (MWM) method



## 1. Introduction

The implementation of information technology in the transportation system has led to an increase in its complexity, resulting in several difficult challenges, such as traffic congestion, which leads to significant economic losses and hidden risks [1]. Therefore, to improve the management of complex transportation systems, research has been focused on intelligent transportation systems that incorporate various technologies, including information integration, data communication, and artificial intelligence [2]. The core of an intelligent transportation system lies in intelligent traffic cognition, which involves predicting traffic flow, an effective approach for enhancing traffic system efficiency and resolving traffic congestion [3]. Accurate traffic flow prediction can aid urban traffic planning and emergency management and mitigate traffic congestion and related issues [4]. Furthermore, traffic flow prediction plays a crucial role in smart city planning, including reducing environmental pollution, shortening travel time, increasing road capacity, and other aspects [5].

The goal of traffic flow prediction is to estimate future traffic conditions of a transportation network based on historical observations. The predicted time span ranges from a few minutes to a few hours. However, the multiple and complex properties of traffic flow data, such as spatial-temporal dependence, external data dependence, high noise and randomness [6,7], pose a very challenging problem. Existing approaches fall into two main categories: knowledge-driven approaches and data-driven approaches. Knowledge-driven

methods usually attempt to establish traffic network modeling through queuing theory and simulated driver behavior in traffic [8,9]. However, this method is now rarely used, because the complexity of traffic flow data is so high that it is difficult for knowledge-driven methods to extract the deep information embedded in the data and the experimental accuracy is often unsatisfactory. With the advent of the information age, available data has shown explosive growth. Data-driven models have dominated this area, shifting from early statistical-based models to recent deep-learning-based models. Benefiting from this trend, prediction accuracy has reached unprecedented heights, and there is still much room for improvement.

Although many data-driven models, such as Deep Spatiotemporal Network (DeepST) [10], Spatiotemporal Residual Network (ST-ResNet) [11], Attention Convolutional Long Short-Term Memory Network (AttoConvLSTM) [12], Diffusion Convolutional Recurrent Neural Network (DCRNN) [6], Attentive Crowd Flow Machine (ACFM) [13] and Attentive Spatio-Temporal Inception ResNet (ASTIR) [14], have achieved excellent results in traffic flow prediction problems, the interpretability of these models is still an open issue, because these models are based on deep neural networks, which have "black box" characteristics. Although these models have superior performance, they often come at the cost of high model complexity, and their results cannot be easily explained or understood by humans [15]. When traffic flow prediction is applied to safety-related fields, such as emergency management, the consequences of prediction errors can be catastrophic; therefore, the interpretability of the model is a critical factor to be considered.

As we are aware, traffic flow prediction models that possess high interpretability could aid in comprehending the decision-making process of the model. On one hand, from a user's perspective, this enhances the persuasiveness and trustworthiness of prediction models. On the other hand, from the perspective of corporations and governments, this allows prediction models to be easily regulated to mitigate ethical issues such as fairness and privacy. However, there is a trade-off between accuracy and interpretability, since high-precision models such as deep neural networks often have complex inference processes, which makes obtaining model interpretability more difficult.

Improving the interpretability of the model while ensuring the accuracy of the traffic flow prediction model is a great challenge. With the advancement of technology, many studies have been proposed to try different methods for interpreting deep neural networks [16,17], among which fuzzy neural networks (FNN) stand out due to the high interpretability of their IF-THEN fuzzy rules [18]. Overall, FNNs can be interpreted by analyzing the internal structure and parameters of the model, indicating that they are inherently transparent, and their results can be fully understood.

Based on the preceding discussion, this paper investigates a solution using fuzzy neural networks for traffic flow prediction and proposes a modified Wang–Mendel method to extract IF-THEN fuzzy rules. The modified method not only achieves superior performance on the BikeNYC dataset, but also precisely tracks errors.

This work addresses the following three research questions: (i) How can fuzzy neural networks be used to build traffic flow prediction models? (ii) How can the Wang–Mendel method be enhanced based on regional variations in data? (iii) How can IF-THEN fuzzy rules improve model interpretability and error tracking accuracy?

To address these issues, we introduce a novel fuzzy-based approach for the traffic flow prediction model by combining fuzzy inference mechanisms and deep neural networks. Building on Wang [19] and Wang and Chen's work [20], we design a novel self-constructing deep fuzzy neural network model that leverages multiple traffic flow data characteristics and has good anti-noise ability. Additionally, it is noteworthy that the SCDFNN exhibits a high degree of interpretability, enabling accurate tracking of model errors by examining the IF-THEN rules. We also propose a modified Wang–Mendel method to enable rapid training of the SCDFNN. Our contributions are summarized as follows.

- **Neuro-Fuzzy Hierarchical Architecture.** We propose SCDFNN for model level traffic data cognitive explanation. To our best knowledge, this is the first neuro-fuzzy

hierarchical architecture that learns logical rules for traffic data cognitive systems without relying on external resources.

- **Modified Wang–Mendel Method.** Considering the difference of data in different regions, we propose a modified Wang–Mendel method. Compared with the ordinary training method, its training model not only has fewer rules and faster training speed but also retains traffic prediction results' interpretability.
- **Comprehensive Experiment.** Comprehensive experimental comparisons and analysis using two public and real datasets demonstrate the superiority of SCDFNN by its high accuracy and high interpretability.
- **Interpretable Rules.** Thanks to SCDFNN, we can discover interpretable traffic flow cognitive fuzzy-based rule sets for given datasets, the former of which have multiple uses.

The structure of this paper is as follows. In Section 2, we provide a review of prior research on traffic flow prediction and fuzzy neural network development. Section 3 covers the problem definition for traffic flow prediction and provides implementation details of our modified Wang–Mendel method. In Section 4, we detail our data processing procedure. Next, in Section 5, we present the design and training methods of the SCDFNN model. Section 6 presents the experimental results. Finally, we conclude our findings in Section 7.

## 2. Related Work

In recent years, the explosion of data in the information age has led to the rapid development of deep neural networks. Numerous excellent deep neural network models have emerged for traffic flow prediction. However, the "black box" property of these models has caused many accidents, leading to concerns about their interpretability. As a result, fuzzy systems, known for their high interpretability, have garnered increasing attention. To improve the interpretability of deep neural networks, some researchers have attempted to combine neural networks with fuzzy systems to form fuzzy neural networks, efforts which have shown promising results [21]. In this section, we will discuss the research progress of traffic flow prediction models and fuzzy neural network models, respectively.

### 2.1. Research Progress of Traffic Flow Prediction

Extensive research has been conducted by numerous scholars on traffic flow prediction using deep neural networks. Two approaches are commonly used for traffic flow prediction: knowledge-driven and data-driven. Knowledge-driven approaches typically model traffic network through queuing theory and simulated driver behavior in traffic. However, these methods struggle to extract deep-level information from complex traffic flow data, resulting in less accurate predictions. In contrast, data-driven methods have become more popular due to the explosion of data in the information age. Recent studies on traffic flow prediction have mostly used data-driven methods [22].

Data-driven methods can be categorized into two types: classical statistical models and machine learning models, particularly deep learning network models, which have shown better performance in terms of accuracy. Classical statistical models include autoregressive integral moving average (ARIMA) [23], seasonal autoregressive integral moving average (SARIMA) [24], vector autoregression (VAR) [25], etc. In contrast, machine learning models, including deep-learning network models, are more effective in capturing the underlying patterns in data. This paper will focus on the second type of model, which will be discussed in detail below.

The Deep Brief Network (DBN) is a multitask model introduced into traffic research by Huang et al. [26]. The researchers added a multi-task regression layer on top of the DBN to predict traffic flow. Chen et al. developed a deep stacked denoising autoencoder model to learn the effects of human movement dependencies on traffic flows by stacking the autoencoder [27]. Yi et al. proposed a deep learning neural network based on TensorFlow and achieved 99% accuracy in traffic flow prediction [28]. Zhang et al. proposed DeepST, a deep-learning-based model for the prediction of spatial-temporal data, and applied it to traffic flow prediction [10]. In 2017, Zhang et al. proposed the ST-ResNet model, which used

the residual neural network framework to model the temporal closeness, period, and trend properties of crowd traffic [11]. Liu et al. proposed the Attention Crowd Flow Machine (ACFM), which used two progressive ConvLSTM units to form a unified neural network module [13]. Zhou et al. proposed an encoder-decoder framework based on convolutional and ConvLSTM units to identify complex features, incorporated a novel attention model AttConvLSTM to emphasize the effects of latent citywide mobility regularities, and applied it to predict Multi-step Citywide Passenger Demands [12]. Yao et al. proposed a Deep Multi-View Spatial-Temporal Network (DMVST-Net) for taxi demand prediction, which used views to model both spatial and temporal relations [29]. Mourad et al. proposed the Attentive Spatio-Temporal Inception ResNet (ASTIR) for traffic flow prediction by combining the Inception–ResNet structure with Convolution-LSTM layers and attention module, which can capture short-term, long-term, period properties, and external factors as well as better capture pattern movement changes [14]. Du et al. proposed DST-ICRL, a deep irregular convolutional residual LSTM network model for urban traffic passenger flow prediction, one which integrates irregular convolutional residential networks and LSTM units to learn spatiotemporal feature representations and fuses external factors [30]. Zhou et al. proposed a filter attention-based spatiotemporal neural network (FASTNN), which used a 3D convolutional neural network to extract universal spatiotemporal dependencies from three types of historical traffic flow and constructed a filtering spatial attention module [31].

In recent years, graph convolutional neural networks (GCN) have emerged as one of the most promising frontiers in deep neural network research, showing superior performance across a variety of application scenarios [32]. To tackle the traffic flow prediction problem, researchers have constructed a non-Euclidean graph input for GCNs by considering road intersections as nodes and road connections as edges. Several GCN-based neural network models have been derived, such as the Spatio-Temporal Graph Convolutional Network (STGCN) and the Attention-based Spatial-Temporal Graph Convolutional Network (ASTGCN), which have achieved outstanding results [33,34].

### 2.2. Development of Fuzzy Neural Networks

Fuzzy neural networks are rooted in fuzzy logic, which was introduced by Zadeh in 1965 [35] to address the imprecision of complex problems. Fuzzy logic replaces the classical Boolean value with a membership degree between 0 and 1 to deal with problems that classical logic cannot represent.

In 1988, Zadeh proposed fuzzy inference systems (FIS) [36], which has since attracted attention from many disciplines over the past three decades [37]. Peng, F. applied FIS to the traffic field and proposed an efficient road traffic anti-collision warning system [38]. Although FIS could offer a more comprehensive understanding of how and why results are generated, it may have to be simplified to meet the transparency requirement. Therefore, FIS's performance falls short compared to deep-learning-based models. Recently, there has been growing research interest in combining the advantages of FIS and neural networks, leading to a new model called the fuzzy neural network model (FNN). Due to its high interpretability and excellent performance, FNN has been widely used in various application scenarios, such as fuzzy clustering [39], nonlinear dynamic system modeling, fault detection and stock prediction [18]. Miao et al. proposed a novel substation-based fire early warning scheme based on fuzzy theory [40]. Huynh et al. proposed a new hybrid algorithm for multi-objective optimum design, combining the Grey Taguchi method, the fuzzy logic system, and an adaptive neuro-fuzzy inference system (ANFIS) algorithm [41]. In this study, we apply FNN to the intelligent traffic cognition field and propose a new traffic flow prediction model.

As we know, one of the reasons that restrict the development of FNN is that the number of rules of FNN will explode when faced with complex problems. To address this issue, researchers have proposed various algorithms, of which the pruning algorithm is the most commonly used. For example, G. Leng et al. proposed a self-organizing fuzzy

neural network (SOFNN) that used adding and pruning techniques based on the geometric growing criterion and recursive least square algorithm to extract the fuzzy rule from input-output data [42]. N. Wang et al. introduced the parsimonious fuzzy neural network (FAOS-PFNN) which utilized growing and pruning of rule-based error criteria and distance criteria, and its parameters were updated by an extended Kalman filter (EKF) method [43]. J. J. Rubio suggested the online self-organizing fuzzy modified least-squares (SOFMLS) method [44], which generated the new rule based on distance criteria and was capable of adding or pruning a neuron from the network in every step of the structure-adjusting strategy. H. Han et al. proposed the GP-FNN model, which calculated the significance of fuzzy rules through Fourier decomposition of the variance of the network output [45]. Simulation studies showed that this model has greater generalization capability and compact and high-performing fuzzy rules. Although the pruning algorithm can reduce the number of fuzzy rules, the setting of the pruning threshold remains challenging. A pruning threshold that is too large or too small can significantly decrease the model's accuracy.

Raju et al. proposed a hierarchical fuzzy system that decomposes a high-dimensional fuzzy system into several layers of low-dimensional fuzzy systems, which reduces the number of rules without using pruning [46]. Wang in 1998 proved that hierarchical fuzzy systems have some discrete properties such as universal approximation [47]. Thereafter, hierarchical fuzzy systems have been widely used in various fields, such as adaptive control [48], multi-objective optimization [49], interpretability [50], and classification [51,52]. In 2020, Wang introduced the deep convolutional fuzzy system (DCFS), which imitates the convolution operator of deep convolutional neural networks (DCNNs) to decompose the high-dimensional fuzzy system into a multi-layer low-dimensional fuzzy system [19]. The Wang–Mendel method is then used to determine the parameters of the fuzzy system layer-by-layer from the bottom-up. The Wang–Mendel (WM) method was first proposed by Wang and Mendel in 1992 and refined by Wang in 2003 [53,54]. It is a method for quickly generating fuzzy rules from data. DCFS has been proven effective in forecasting chaotic random time series and the Heng Seng index (HSI) of the Hong Kong stock market. However, the Wang–Mendel method used in DCFS divides membership functions evenly without considering the regional differences in input data.

## 3. Preliminaries

In this section, we provide a detailed explanation of the structure and operational mechanism of the FNN, along with instructions on how to build it using the modified Wang–Mendel method. Furthermore, we conclude this chapter by defining the traffic flow prediction problem. Table 1 lists the key symbols used in this article.

**Table 1.** List of Symbols.

| Symbol Name | Significance |
| --- | --- |
| $N$ | Number of samples |
| $x_i$ | $i$-th dimension input, $i = 1, 2, \ldots, n$ |
| $y$ | Sample output |
| $m_i$ | Initial score of $i$-th dimensional membership function |
| $MSE_l(y)$ | the variance of output $y$ corresponding to the sample in each region, $l = 1, 2 \ldots m_i - 1$ |
| $q_i$ | Number of membership functions in the $i$-th dimension after pruning |
| $\mu_{j_i}$ | The $j$-th membership function corresponding to the $i$-th dimension input, $j_i = 1, 2, \ldots, q_i$ |
| $A_i^{j_i}$ | Fuzzy semantics corresponding to $\mu_{j_i}$ |
| $c^{j_1, j_2, \ldots, j_n}$ | Rule parameters of fuzzy rule $(j_1, j_2, \ldots, j_n)$ |
| $f_k^{j_1, j_2, \ldots, j_n}$ | Activation strength of the $k$-th sample on rule $(j_1, j_2, \ldots, j_n)$ |
| $X_t$ | Inflow and outflow matrix at time $t$, $X_t \in \mathbb{R}^{2 \times I \times J}$ |
| $P_{i, j, t}$ | Set of vehicles with time $t$ grid $(i, j)$ |
| $card$ | Finding the total number of elements in a set |
| $x_{i,j,t}^{in/out}$ | Inflow/outflow in $(i, j)$ grid at time $t$ |
| $E_t$ | External data at time $t$ |
| $\tau$ | Prediction time |

### 3.1. The Structure and Working Mechanism of the FNN

The FNN is a hybrid system composed of artificial neural networks and fuzzy systems, where the former provides learning ability [55], and the latter provides interpretability and handles uncertainty. Typically, the structure of the FNN is divided into layers, with each layer responsible for executing specific tasks; it can be broadly categorized into two parts: layers that implement the premise part of the rule, and layers that implement the consequent part of the rule [37]. While the number of layers in the FNN design can vary in specific implementations, the functions implemented are the same. In this section, we introduce a commonly-used four-layer structure model known as the IMRO structure, as illustrated in Figure 1.

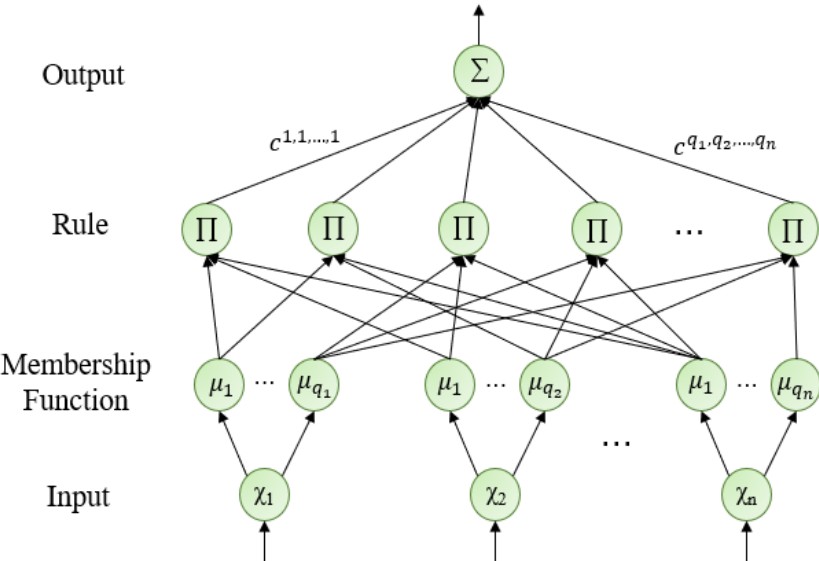

**Figure 1.** IMRO structure of FNN.

The IMRO structure of FNN is divided into four layers: input layer, membership function layer, rule layer and output layer. In the input layer, the raw input is fed to the next layer without any manipulation. In the membership-function layer, each input $x_i$ is input into $q_i$ different membership functions. Each membership function represents a kind of semantics, and its output is the membership degree that the input $x_i$ satisfies this semantics, and its value is between 0 and 1. In the rule layer, each rule node selects a membership function $q_{j_i}$ from the membership function corresponding to each input, which constitutes the precondition of a fuzzy rule, and each rule node also corresponds to a rule parameter $c^{j_1, j_2, \ldots, j_n}$. The rule corresponding to this rule node is:

$$IF\ x_1\ is\ A_1^{j_1}\ and\ x_2\ is\ A_2^{j_2} \ldots and\ x_n\ is\ A_n^{j_n},\ THEN\ y\ is\ c^{j_1, j_2, \ldots, j_n}, \tag{1}$$

where $x_i$ is the input of FNN, $i = 1, 2, \ldots, n$; $A_i^{j_i}$ represents a kind of fuzzy semantics, such as "fast", "slow", "high" or "low" that cannot be accurately expressed, which is mathematically represented as a membership function, $j_i = 1, 2, \ldots, q_i$; the rule parameter $c^{j_1, j_2, \ldots, j_n}$ is a precise value.

The output of the rule node is the activation strength $f_k^{j_1, j_2, \ldots, j_n}$ of the rule, and the formula for calculating $f_k^{j_1, j_2, \ldots, j_n}$ is:

$$f_k^{j_1, j_2, \ldots, j_n} = \prod_{i=1}^{n} \mu_{j_i}(x_i(k)). \tag{2}$$

In this formula, $\mu_{j_i}$ represents the membership function that represents the fuzzy semantic $A_i^{j_i}$. The expression $\mu_{j_i}(x_i(k))$ means the membership degree of the $k$-th sample for the fuzzy proposition "$x_i$ is $A_i^{j_i}$". The symbol $\prod$ represents the fuzzy intersection operator, which

means "*and*". Therefore, the activation strength $f_k^{j_1, j_2, \ldots, j_n}$ represents the degree to which the input data meets the preconditions of the rule "*IF $x_1$ is $A_1^{j_1}$ and $x_2$ is $A_2^{j_2}$ … and $x_n$ is $A_n^{j_n}$*".

Finally, in the output layer, we accumulate the outputs of each rule to obtain the overall output $y$ of FNN, where the output of each rule is the activation strength $f_k^{j_1, j_2, \ldots, j_n}$ of the rule multiplied by the rule parameter $c^{j_1, j_2, \ldots, j_n}$. The calculation formula for $y$ is:

$$y = \sum_{j_1=1}^{q_1} \cdots \sum_{j_n=1}^{q_n} c^{j_1, j_2, \ldots, j_n} \times f_k^{j_1, j_2, \ldots, j_n}. \tag{3}$$

*3.2. Modified Wang–Mendel Method*

In 1992, Wang and Mendel proposed the Wang–Mendel (WM) method for extracting rules from data to form a rule base for fuzzy systems [53]. Wang further improved the method in 2003 [54]. The WM method has the advantage of training the fuzzy system parameters using only one pass through the training data, making it faster than iterative algorithms. However, it faces difficulties with high-dimensional problems due to rule explosion. In 2020, Wang proposed the deep convolutional fuzzy system (DCFS) model to address this issue by designing the low-dimensional fuzzy systems in a bottom-up, layer-by-layer fashion using the Wang–Mendel method. However, the membership function generated by the Wang–Mendel method in DCFS does not consider the regional differences in data, leading to many redundant rules and increased training time. To address this, we modified the membership function division of the Wang–Mendel method to generate uneven rules. The modified Wang–Mendel method is presented below.

**Step 1:** Given $N$ training input-output data pairs, each data pair is denoted as:

$$[x_1(k), \ x_2(k), \ldots, x_n(k); y(k)], \tag{4}$$

where $x_i(k)$ is the input of data, $y(k)$ is the output, $k \in 1, 2, \ldots, N$, $i \in 1, 2, \ldots, n$.

Determine a preset number of membership functions for each input dimension $x_i$, expressed as $m_i$. The $m_i$ is related to the physical meaning of the input dimension. Specifically, the more complex the input dimension, the larger the $m_i$ should be set.

**Step 2:** Calculate the maximum value $\max x_i$ and minimum value $\min x_i$ of each dimensional variable x of the input data, and divide the region $[\min x_i, \ \max x_i]$ into $m_i - 1$ blocks:

$$[x_1, \ x_2], [x_2, \ x_3], \ldots, [x_{m_i-1}, x_{m_i}], \tag{5}$$

where $x_1 = \min x_i$, $x_{m_i} = \max x_i$. Calculate the variance of output $y(k)$ corresponding to the sample in each region $[x_l, \ x_{l+1}]$ in (5) and record it as $MSE_l(y)$, where $l = 1, 2 \ldots m_i - 1$. Note:

$$MSE\prime(y) = \min MSE_l(y) + d \times (\max MSE_l(y) - \min MSE_l(y)), \tag{6}$$

where $\max MSE_l(y)$ and $\min MSE_l(y)$ are the maximum and minimum values in $MSE_l(y)$, respectively, and $d$ is a hyperparameter, $0 < d < 1$. The $MSE\prime(y)$ will be used as a pruning related threshold in subsequent steps.

**Step 3:** For the dimension whose partition number $m_i$ satisfies $m_i > m\prime$, we perform a pruning operation, where $m\prime$ is a hyperparameter. We traverse all regions from the first region $[x_1, \ x_2]$. If the current traversal region is $[x_a, x_b]$, and the region $[x_b, x_c]$ has not been traversed, and satisfy:

$$MSE_{ab}(y) < MSE\prime(y), \tag{7}$$

$$MSE_{bc}(y) < MSE\prime(y), \tag{8}$$

$$x_c - x_a \le z\prime \times \frac{\max x_i - \min x_i}{m_i}. \tag{9}$$

Then merge region $[x_a, x_b]$ and region $[x_b, x_c]$ into region $[x_a, x_c]$ as

$$MSE_{ac}(y) = \frac{MSE_{ab}(y) + MSE_{bc}(y)}{2},$$ (10)

where $MSE_{ab}(y)$, $MSE_{bc}(y)$ and $MSE_{ac}(y)$ are the variances of the output of corresponding regions $[x_a, x_b]$, $[x_b, x_c]$ and $[x_a, x_c]$, and $z\prime$ is a hyperparameter. The merged region must be less than $z\prime$ times of the original region. After region merging, we will check whether $[x_a, x_c]$ can be merged with the next region $[x_c, x_d]$. If not, we will traverse the next region until all regions are traversed.

**Step 4:** For each merged region, we take the center of the region as the vertex of the membership function, and the center of the adjacent region as the endpoint of the membership function to construct a triangle membership function. For the two regions on the edge, the left and right endpoints of their corresponding membership functions are $-\infty$ and $+\infty$, respectively. Then, we will obtain $q_i$ membership functions $\mu_{j_i}(x)(\,j_i = 1, 2, \ldots, q_i)$; each membership function represents a semantic $A_i^{j_i}$. Figure 2 shows the trigonometric membership function cluster obtained by merging some inputs.

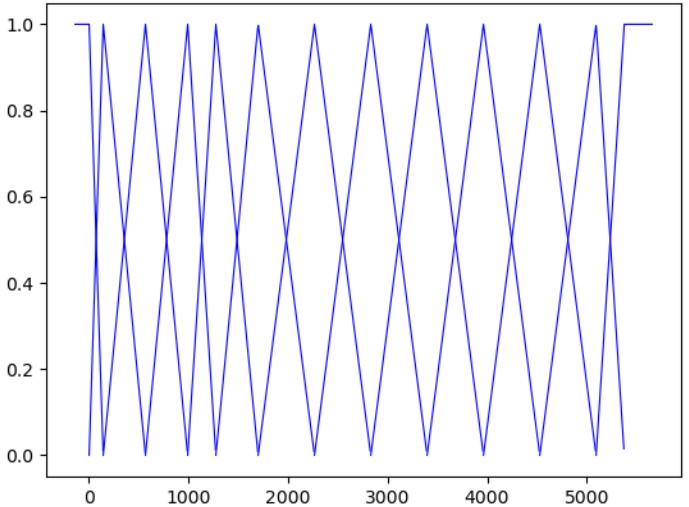

**Figure 2.** Triangular membership function clusters obtained after pruning.

**Step 5:** Generate a rule space, which contains $\prod_{i=1}^{n} q_i$ rule parameters $c^{j_1, j_2, \ldots, j_n}$, and the rule corresponding to each rule parameter is shown in (1).

**Step 6:** Traverse all training data, and for each training data $[x_1(k), x_2(k), \ldots, x_n(k); y(k)]$, calculate the activation strength $f_k^{j_1, j_2, \ldots, j_n}$ of each rule according to Equation (2), and save the maximum activation strength $f_k^{j_1^*, j_2^*, \ldots, j_n^*}$ and output $y_k$. The formula for calculating $f_k^{j_1^*, j_2^*, \ldots, j_n^*}$ is

$$f_k^{j_1^*, j_2^*, \ldots, j_n^*} = \max_{j_1, j_2, \ldots, j_n} f_k^{j_1, j_2, \ldots, j_n}.$$ (11)

After traversing all $N$ training samples, if a rule has saved at least one $\left\{ f_k^{j_1^*, j_2^*, \ldots, j_n^*}, y_k \right\}$ data pair, the rule parameter $c^{j_1^*, j_2^*, \ldots, j_n^*}$ of this rule can be calculated according to the following equation:

$$c^{j_1^*, j_2^*, \ldots, j_n^*} = \frac{\sum_{k=1}^{N} f_k^{j_1^*, j_2^*, \ldots, j_n^*} \times y_k}{\sum_{k=1}^{N} f_k^{j_1^*, j_2^*, \ldots, j_n^*}},$$ (12)

where $N$ is the total number of training samples, $(j_1^*, j_2^*, \ldots, j_n^*)$ represents the coordinates of this rule in the rule space, and $f_k^{j_1^*, j_2^*, \ldots, j_n^*}$ is the activation intensity of the $k$-th sample on

the rule, and is the maximum activation intensity among all rules triggered by the *k*-th sample. $y_k$ is the output of the *k*-th sample.

**Step 7:** For a rule whose rule parameter is not calculated in Step 6, we will obtain its rule parameter from its neighbor rules. Specifically, its rule parameter is the arithmetic mean of the rule parameters of the neighbor rules with rule parameters. Repeat this step until all rules in the rule space have a rule parameter. The sufficient and necessary condition for rule $(j_1, j_2, \ldots, j_n)$ and $(j'_1, j'_2, \ldots, j'_n)$ to be neighbors is that there is and only one $r \in 1, 2, \ldots, n$, so that $j_r = j'_r + 1$ or $j_r = j'_r - 1$.

For the trained FNN, we use Equation (3) to obtain its output *y*.

In contrast to the ordinary Wang–Mendel method, our proposed improvement involves adding pruning operations consisting of Steps 2, 3 and 4. The ordinary Wang–Mendel method employs triangle membership functions to uniformly cover the data, without considering the regional differences within the data. In our approach, we first partition the input data into uniform regions, and then evaluate the influence of each region using variance analysis. We combine adjacent regions with low influence to form larger regions, for which we then obtain a membership function. This method reduces the number of rules and accelerates the training speed. Our experiments show that the modified Wang–Mendel method reduces the training time by more than half compared to the ordinary Wang–Mendel method. The flowchart of the Modified Wang–Mendel Method is presented in Figure 3.

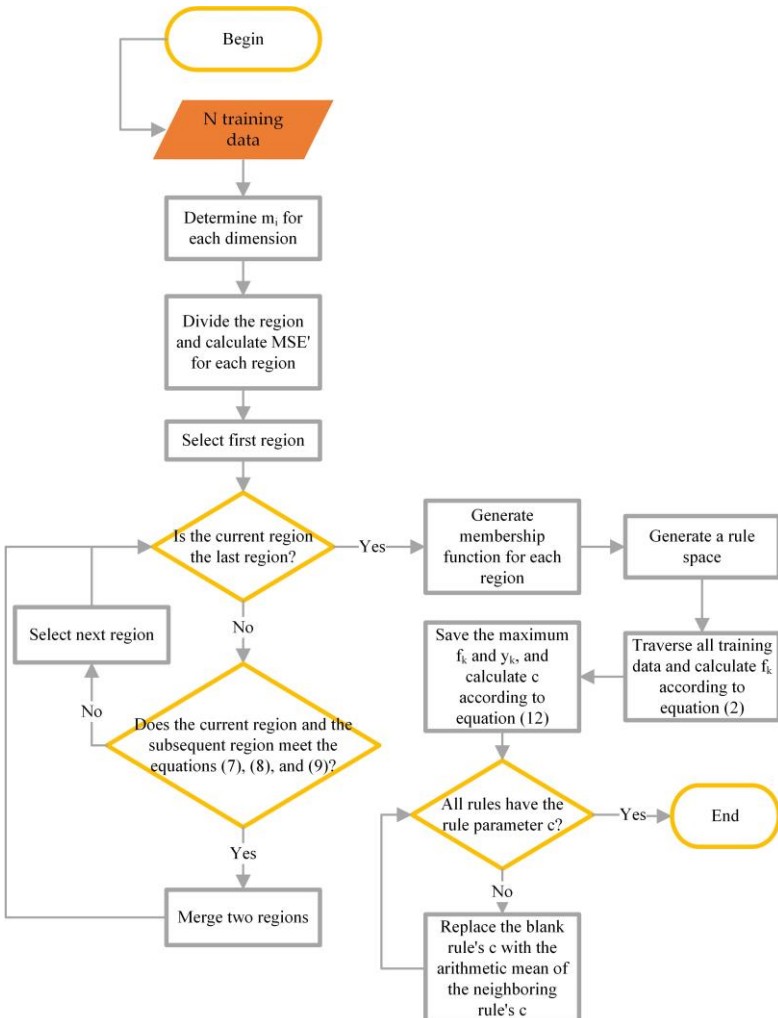

**Figure 3.** The flowchart of the Modified Wang–Mendel Method.

### 3.3. Problem Definition

In traffic flow prediction, two map representations are commonly used: the topology map structure and the grid map structure. The topology map structure represents intersections as nodes, and the roads as edges, making it suitable for graph convolution (GCN) to capture spatial information. The grid map structure, on the other hand, divides the map into equally-sized areas based on longitude and latitude, with each area considered as a research object. After a comparative analysis, we found that the grid map structure is more appropriate for our study. We divided the map into $I \times J$ grid map using longitude and latitude information, as illustrated in Figure 4.

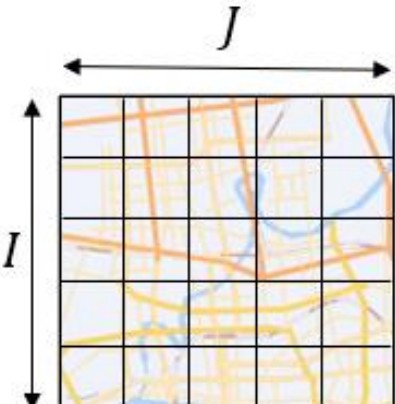

**Figure 4.** Map divided into an $I \times J$ grid.

Let the set of vehicles contained in the $(i, j)$ grid at time $t$ be $P_{i, j, t}$, then the inflow of the $(i, j)$ grid at time $t$ is $x_{i,j,t}^{in} = card(\{g | g \notin P_{i, j, t-1} \wedge g \in P_{i, j, t}\})$. Similarly, the outflow of the $(i, j)$ grid at time $t$ is $x_{i,j,t}^{out} = card(\{g | g \in P_{i, j, t-1} \wedge g \notin P_{i, j, t}\})$.

In addition, we use $X_t \in \mathbb{R}^{2 \times I \times J}$ to represent the inflow and outflow at time $t$, and $E_t$ to represent the external factors at time $t$, where $E_t$ includes the weather, weekend, holiday and time location at time $t$. Then the traffic flow prediction problem can be described by Definition 1.

**Definition 1.** *Given historical values* $\{X_t, E_t | t = 0, \ldots, \tau - 1\}$, *Predict* $X_\tau$.

Because fuzzy neural network is suitable for single output model, we decompose the problem described by Definition 1 into $2 \times I \times J$ single output prediction tasks described by Definition 2.

**Definition 2.** *Given historical values* $\{X_t, E_t | t = 0, \ldots, \tau - 1\}$, *predict* $x_{i,j,\tau}^{in/out}$.

## 4. Data Processing

The characteristics of traffic flow data are diverse and complex, which presents significant challenges for traffic flow prediction. In this section, we will provide details on how to process raw data to extract spatial-temporal knowledge that is information-rich and representative, as well as contextual knowledge that includes spatial-temporal dependence, periodicity, and external data dependence.

### 4.1. Spatial-Temporal Dependence

For the inflow $x_{i,j,t}^{in}$ of the location $(i, j)$ at time $t$, we calculate the sum of the outflow from its neighbors in the previous $s$ time slots and that of distant neighbors, denoted as $neighbor1_{i,j,t}^{out}$, $neighbor2_{i,j,t}^{out}$ respectively, so that to reflect both temporal and spatial

dependence of the data. The intuitive is based on the observation that the inflow of a location at a certain time slot usually comes from the outflow from its surrounding locations, and vice versa for the outflow. The specific calculation formula for $neighbor1^{out}_{i,j,t}$ and $neighbor2^{out}_{i,j,t}$ is:

$$neighbor1^{out}_{i,j,t} = \sum x^{out}_{i^{neighbor1}, j^{neighbor1}, t'} \tag{13}$$

$$neighbor2^{out}_{i,j,t} = \sum x^{out}_{i^{neighbor2}, j^{neighbor2}, t}. \tag{14}$$

where $\left(i^{neighbor1}, j^{neighbor1}\right)$ is the eight adjacent grids of $(i, j)$, and $\left(i^{neighbor2}, j^{neighbor2}\right)$ is the sixteen grids separated by one grid from $(i, j)$, as shown in Figure 5. If the $\left(i^{neighbor1}, j^{neighbor1}\right)$ or the $\left(i^{neighbor2}, j^{neighbor2}\right)$ grid exceeds the map boundary, we set the inflow or outflow corresponding to the grid to 0.

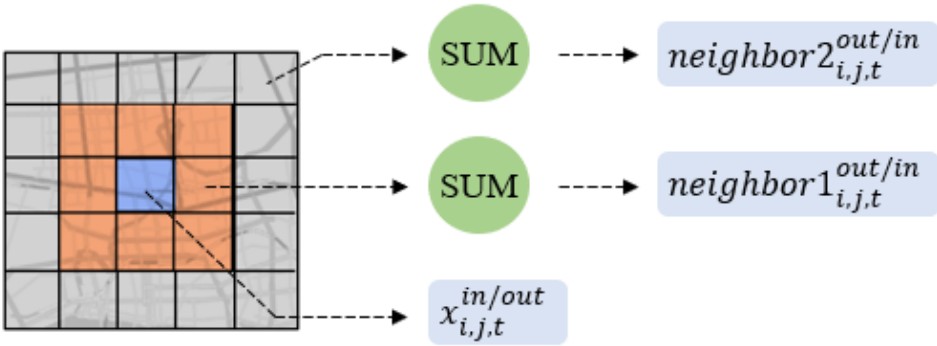

**Figure 5.** Spatial-temporal dependent data extracted from maps.

*4.2. Periodicity*

Traffic flow data, as a typical time-series data, is naturally dependent on adjacent time slots, known as temporal closeness in the literature. Moreover, traffic flow data also exhibits a clear daily and weekly periodicity, generally attributed to regular human mobility. Therefore, models that consider all three types of temporal factors consistently tend to demonstrate superior performance. In this regard, we develop three types of periodicity data, namely, closeness, period, and trend, to reveal the periodicity characteristics of traffic flow data. Figure 6 illustrates these periodicity data types.

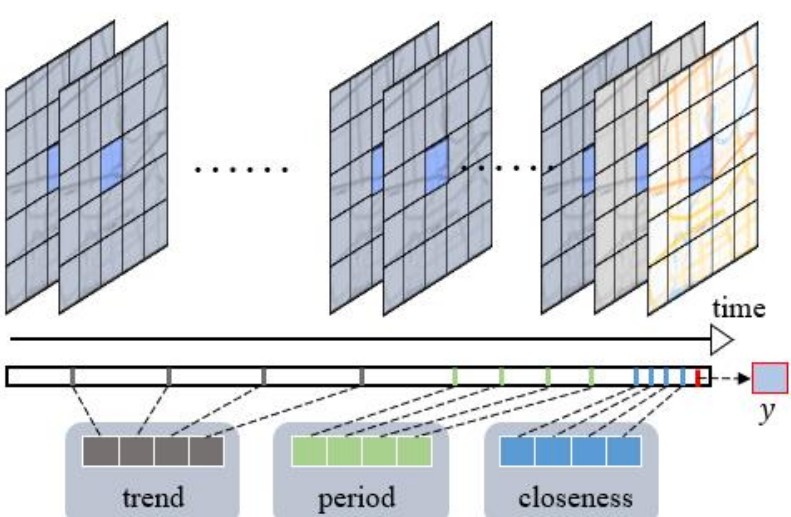

**Figure 6.** Periodicity of a certain location.

For the missing values of period and trend, we fill them with the data of time slot $t − 1$.

### 4.3. External Data Dependence

In addition to spatial and temporal factors, various external factors may influence traffic patterns, such as weather, holidays, and temporal location. To investigate the impact of external factors on traffic flow, we visualized the inflows and outflows at coordinates (11, 23) in the TaxiBJ dataset during February 2016, as shown in Figure 7a. Clearly, the significant decrease in traffic flow during holidays confirms our hypothesis. Additionally, we created a flow heatmap at a specific time on 13 February 2016, as depicted in Figure 7b. The clustering characteristics of high flow areas indicate that neighboring regions significantly affect traffic flow.

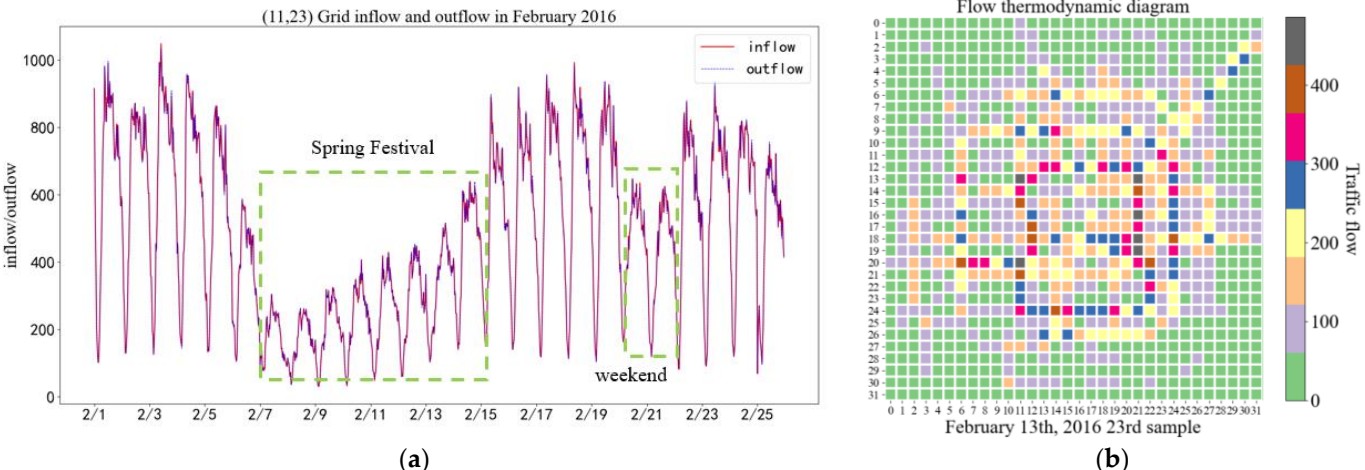

**Figure 7.** (**a**) Visualization inflow and outflow in February 2016; (**b**) flow heatmap on 13 February 2016.

We use $isH_t$ to indicate whether time $t$ is a holiday, and $isW_t$ to indicate whether time $t$ is a weekend. Specifically, when the corresponding time is a holiday, $isH_t$ is set to 1, otherwise 0, as is $isW_t$.

Simultaneously, weather conditions may have a direct impact on human mobility. As the TaxiBJ dataset contains 17 types of weather data, we map them to numbers 0 to 16 and denote the weather at time t as $Weather_t$. Note that the Bike NYC dataset doesn't have weather data.

In addition, we observe that the time of traffic peak is usually fixed in a day and the traffic flow is closely related to the position of the sampled timestamp. Based on this pattern, we use $time_t$ to represent the timestamp position of time t. For the TaxiBJ dataset, there are 48 samples per day and the value of $time_t$ ranges from 1 to 48. For the Bike NYC dataset, there are 24 samples per day and the value of $time_t$ ranges from 1 to 24.

## 5. Our Proposed Approach

Throughout this section, the structure of our proposed SCDFNN model and its learning process are described in detail. The whole framework of the SCDFNN model is shown in Figure 8. Taking the task of predicting the inflow $x_{i,j,n}^{in}$ as an example, we describe each module in the framework in detail in Sections 5.1–5.3. For the task that predicts the outflow $x_{i,j,n}^{out}$, we swap the inflow and outflow, as below.

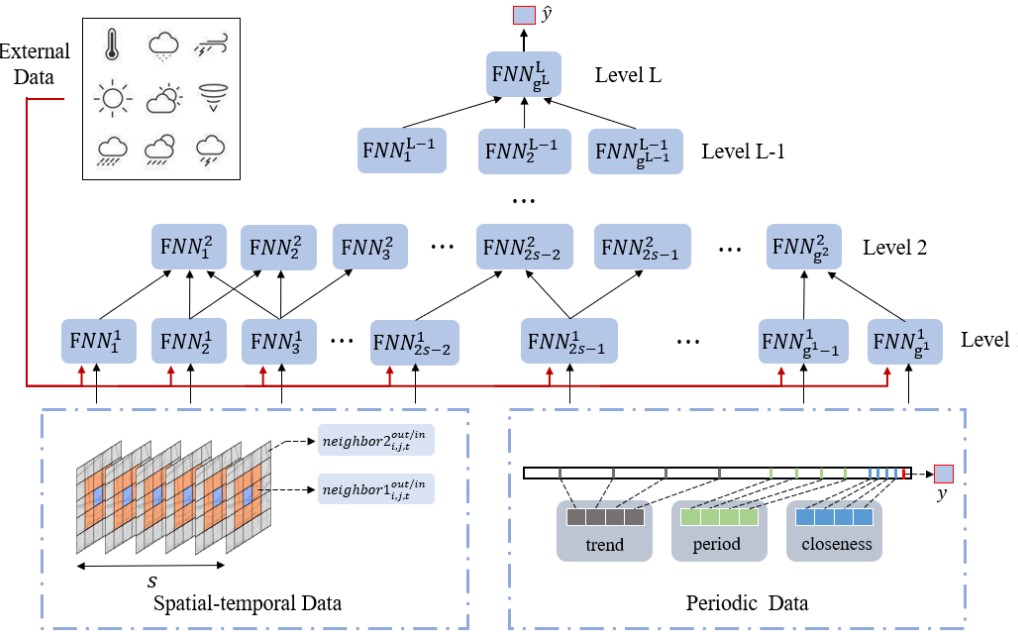

**Figure 8.** Overall framework of SCDFNN.

### 5.1. Spatial-Temporal Data Module

The integral framework of the Spatial-temporal data module is shown in Figure 9. We take the $neighbor1^{out}_{i,j,t}$ and $neighbor2^{out}_{i,j,t}$ of the past $s$ time slots for time $n$ to represent the data spatial-temporal dependence, where $t = \tau - s, \dots, \tau - 1$. We feed $neighbor1^{out}_{i,j,t}$ to the first $s - 1$ FNNs with a sliding window of size 2, and $neighbor2^{out}_{i,j,t}$ to the following $s - 1$ FNNs. Meanwhile, we feed external data that includes $isH_\tau$, $isW_\tau$, $time_\tau$ and $Weather_{\tau-1}$ to each FNN. For the BikeNYC dataset, the external data only includes $isH_\tau$, $isW_\tau$ and $time_\tau$.

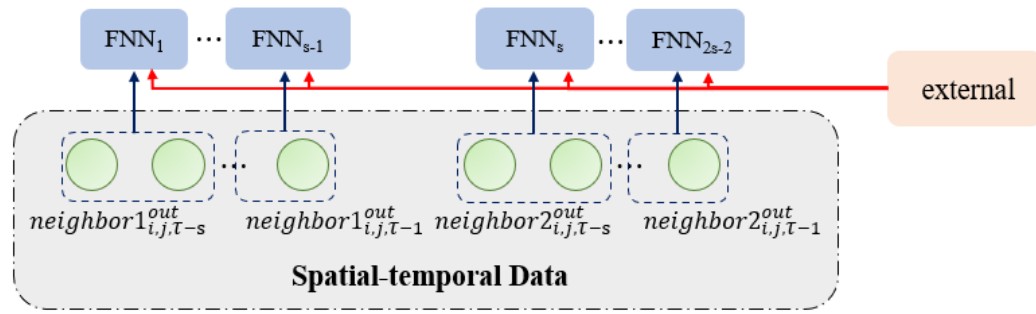

**Figure 9.** Structure of the spatial-temporal module.

### 5.2. Periodic Data Module

The integral framework of the Periodic data module is shown in Figure 10, which combines three useful temporal factors named closeness data, period data, and trend data, respectively. For the closeness data, which includes the inflow data of the recent $h$ time intervals, similar to the spatial dependence module, we use a sliding window of size 2 to map these $c$ inflow data into $h - 1$ FNNs.

Period data includes inflow data of $p$ time slots and the sampling interval is $T_p$. Specifically, $T_p$ is 48 (or 24 in BikeNYC dataset) which is equivalent to one day. Similarly, we map the $p$ inflow data into $p - 1$ FNNs with a sliding window of size 2.

Similarly to periodic data, trend data includes inflow data of $q$ time slots with the sampling interval $T_q$, which is set as one week. We also utilize a sliding window of size 2 to map these $q$ inflow data into $q - 1$ FNNs. It turns out that we achieve the best result when $p$ and $q$ are set to 1, and the training process is the fastest too.

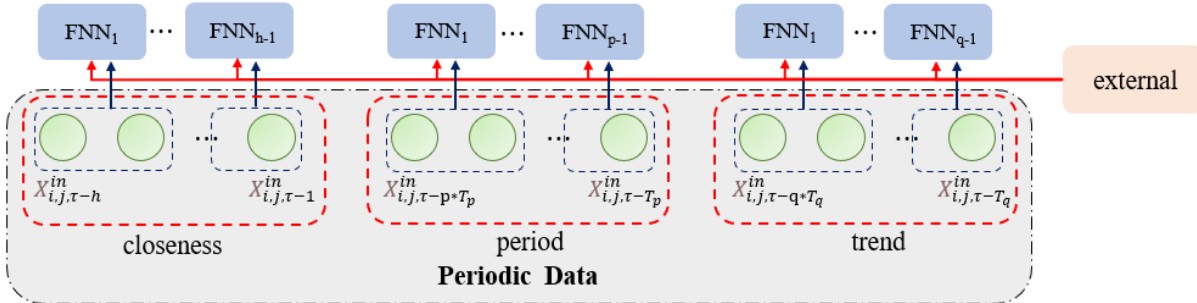

**Figure 10.** Structure of the Periodic Data module.

Additionally, we also add external data to each FNN in this module, including all the external data at time n, that is: $isH_\tau$, $isW_\tau$, $time_\tau$, and $Weather_{\tau-1}$. Meanwhile, we add both $isH_{\tau-p*T_p}, \dots, isH_{\tau-T_p}$ and $isW_{\tau-p*T_p}, \dots, isW_{\tau-T_p}$ to the period data, and $isH_{\tau-q*T_q}, \dots, isH_{\tau-T_q}$ to the trend data.

*5.3. Hierarchical Fuzzy System*

By leveraging the aforementioned two modules, we can obtain the first layer of $g_1$ FNN, where $g_1 = 2s + h + p + q - 5$. We train each FNN with the modified Wang–Mendel method using the data pair $[input; y]$ of the training set, where *input* is the input of each FNN as described before, $y = x^{in}_{i,j,\tau}$. During training, we feed the training set data to the first FNN layer, and the obtained output is then used as the input of the second FNN layer. A sliding window of size 3 is used to map the first FNN layer to the second layer FNN. For the second FNN layer, we also use $y$ as the label to train all the FNNs with the modified Wang–Mendel method. We will repeat this procedure, building the FNNs layer by layer from the bottom up, until the entire SCDFNN is constructed. We take the output of the final FNN in the last layer as the output of the entire model $\hat{y}$.

## 6. Experimental Design and Results

The experimental section of this paper begins by outlining the experimental setup and then presents the results of the experiments in a comprehensive manner. The main objective is to answer the following research questions:

Q1. Can the interpretable traffic flow prediction model SCDFNN achieve prediction accuracy comparable to that of well-performed deep neural network models?

Q2. Did each component of the SCDFNN model contribute positively to the prediction performance?

Q3. What are the advantages of using SCDFNN compared to deep neural network models?

*6.1. Experiment Settings*

In this subsection, we will describe the dataset used, as well as the baselines and evaluation metrics employed.

6.1.1. Data Set

To ensure the reliability of our experiments, we employ two widely used and publicly available datasets, namely, TaxiBJ and BikeNYC.

The TaxiBJ dataset consists of Beijing taxi GPS data, weather data, and holiday data. It covers four time periods: 1 July 2013 to 30 October 2013, 1 March 2014 to 30 June 2014, 1 March 2015 to 30 June 2015, and 1 November 2015 to 10 April 2016. The dataset comprises 22,459 samples, and the time interval between each sample is 30 min. Each sample is accompanied by its corresponding weather condition, which ranges from 0 to 16. We rank the weather conditions based on their impact on traffic, with higher values indicating greater impact. For instance, Sunny is ranked 0, while Dusty is ranked 16. The dataset also includes holiday information for 41 days. We mark the samples corresponding to these

41 days as holiday samples and set the value of $isH_n$ to 1. For this dataset, we use the most recent 4 weeks of data as test data and all remaining data as training data.

On the other hand, the BikeNYC dataset comprises trip records and holiday information for bicycles in New York. Each trip record contains details such as trip duration, origin and destination IDs, and origin and destination time. It only covers one time span of 1 April 2014 to 30 September 2014, and the time interval is one hour. It consists of 4392 available time slots. Although the dataset does not include weather information, it provides holiday data for 20 days, which we label as holiday samples. For this dataset, we choose the last 10 days as test data and all other data as training data.

Table 2 summarizes the essential details of both datasets, such as data type, location, time period, and time interval.

**Table 2.** Dataset used.

| Dataset | TaxiBJ | BikeNYC |
| --- | --- | --- |
| Data type | Taxi GPS | Bike rental |
| Location | Beijing | New York |
| Time span | 7/1/2013–4/10/2016 | 4/1/2014–9/30/2014 |
| Time interval | 30 min | 1 h |
| Gird map size | (32, 32) | (16, 8) |
| Average sampling rate (s) | ~60 | \ |
| Number of taxis/bikes | 34,000+ | 6800+ |
| Available time interval | 22,459 | 4392 |
| Holiday | 41 | 20 |
| Weather conditions | 17 types | \ |

6.1.2. Baselines

In order to demonstrate the superiority of the proposed method in this paper, we compare SCDFNN with 12 widely-used baselines commonly applied in traffic flow prediction problems.

1.  **ARIMA:** The Autoregressive Integrated Moving Average model is a well-known model for analyzing and predicting time series data, and is often used for traffic flow forecasting.
2.  **SARIMA:** The Seasonal Autoregressive Integrated Moving Average model takes into account seasonal properties of data, and is based on the ARIMA model, enabling it to learn tightness and cyclical dependencies outside of ARIMA.
3.  **VAR:** The Vector Autoregressive Model is an advanced spatiotemporal model that captures pairwise relationships between all streams, but it is computationally expensive due to the large number of parameters it uses.
4.  **ST-ANN:** This model extracts spatial and temporal information by taking 8 nearby spatial values and 8 previous time steps, respectively, and feeds the resulting spatiotemporal features into an artificial neural network.
5.  **DeepST:** This model models spatiotemporal data as temporal compactness, cycles, and seasonal trends using a deep neural network-based model that utilizes convolutional layers.
6.  **ST-ResNet:** The Residual network is based on deep convolution, using a traditional square convolution kernel. It is used to capture temporal and spatial dependencies in spatiotemporal data.
7.  **AttoConvLSTM:** This model employs an encoder-decoder framework based on convolutional and attention LSTM to capture spatiotemporal features.
8.  **DMVST-Net:** This model is a spatiotemporal neural network that predicts taxi demand by integrating information from three views, namely, temporal view, spatial view and semantic view.
9.  **DCRNN:** DCRNN is a diffuse convolutional recurrent neural network that uses bidirectional graph random walks to model spatial dependencies and recurrent neural networks to capture temporal dynamics.

10. **DST-ICRL:** This model uses LSTM units and an irregular convolutional residential network to learn spatiotemporal features.
11. **ACFM:** The ACFM model is an attentive-LSTM-based spatiotemporal data model.
12. **ASTIR:** ASTIR combines an attention module, a convolution-LSTM layer, and the Inception-ResNet structure.

6.1.3. Evaluation Metrics

We chose RMSE (Root Mean Squared Error) and MAE (Mean Absolute Error) to evaluate our model, which are commonly used in literature.

$$RMSE = \sqrt{\frac{1}{N_{test}} \sum_{i=1}^{N_{test}} (\hat{y}_i - y_i)^2}. \tag{15}$$

$$MAE = \frac{1}{N_{test}} \sum_{i=1}^{N_{test}} |\hat{y}_i - y_i| \tag{16}$$

where $\hat{y}_i$ is the predicted value of the model, $y_i$ is the ground truth value, and $N_{test}$ is the number of test-set samples.

*6.2. Experimental Results*

This section presents the extensive experimental results of the SCDFNN on the TaxiBJ and BikeNYC datasets to address the three research questions raised earlier. First, we comparatively analyzed the performance of SCDFNN and other baselines using both datasets. We present the parameter settings used for the BikeNYC dataset in Table 3. Next, we conducted ablation experiments and anti-noise experiments on SCDFNN to demonstrate its excellent anti-noise capability. Finally, we analyzed the complexity of the improved Wang–Mendel method in comparison to the original Wang–Mendel method.

**Table 3.** Parameters settings of SCDFNN on the BikeNYC dataset.

| Name | Significance | Value |
|------|-------------|-------|
| $m_{Periodic}$ | The initial number of divisions of Periodic Data Modules | 38 |
| $m_{ST}$ | Same as above, the Spatial-temporal data module | 38 |
| $m_{isW}$, $m_{isH}$ | Same as above, "is weekend" and "is holiday" | 2 |
| $m_{time}$ | Same as the above, the timestamp | 20 |
| $m^L$, $L \geq 2$ | Number of initialization divisions above the second layer | 28 |
| $m'$ | Minimum initial score for pruning | 5 |
| $d$ | Pruning Threshold | 0.8 |
| $h$ | Length of closeness data | 8 |
| $p, q$ | Length of period data or trend data | 1 |
| $s$ | Length of space data | 2 |

6.2.1. Comparison Analysis (Q1)

We compared the SCDFNN model proposed in this paper with baseline models, and the results are shown in Tables 4 and 5. The following results can be observed therein:

- Traditional regression-based models, such as VAR, ARIMA and SARMA, do not perform as well as other models due to the following reasons: (1) VAR is a shallow model and has lower model capacity than deep neural networks. It also lacks the ability to capture external information. (2) ARIMA and SARMA were designed for time series prediction, and are not optimized for traffic flow, thus they lack the mechanism to use spatial dependency and external information.
- Among the deep neural network models, DCRNN, AFCM and ASTIR perform better than DeepST and ST ResNet because they integrate more components, similarly to AttoConvLSTM and AFCM, which utilize convolution and attention LSTM, and ASTIR, which leverages Inception-ResNet, Convolution-LSTM, and attention module, which enable them to capture more information from the data. Thus, it is common

to stack modules or layers to achieve good performance in deep neural networks. However, this also makes the model more complex and difficult to interpret.

- SCDFNN performs better than deep neural networks on the BikeNYC dataset, but its performance on the TaxiBJ dataset is not as expected. This could be due to taxis being faster than bicycles and using overpasses, making it challenging for SCDFNN to capture spatial information on the TaxiBJ dataset. Improving the design of space modules may bring better results for SCDFNN.

- The SCDFNN model outperforms the deep neural network models (i.e., ACFM and ASTIR) on the BikeNYC dataset, indicating its superiority in achieving sufficient accuracy and interpretability. Additionally, utilizing the modified Wang–Mendel method to train the SCDFNN model yields better performance than using the traditional Wang–Mendel method. As discussed later, the modified Wang–Mendel method is significantly faster than the traditional approach.

**Table 4.** Comparison results of different methods on TaxiBJ datasets.

| Method | TaxiBJ | |
|---|---|---|
| | RMSE | MAE |
| ARIMA | 22.78 | 7.25 |
| SARIMA | 26.88 | 8.51 |
| VAR | 22.88 | 7.47 |
| ST-ANN | 19.57 | 6.23 |
| DeepST | 18.18 | 6.21 |
| ST-ResNet | 16.69 | 5.41 |
| AttoConvLSTM | 17.41 | 6.04 |
| DMVST-Net | 15.57 | 5.28 |
| DCRNN | **15.04** | **5.10** |
| ACFM | 15.4 | - |
| SCDFNN (WM) | 17.24 | 11.14 |
| SCDFNN (modified WM) | 16.82 | 10.23 |

**Table 5.** Comparison results of different methods on BikeNYC datasets.

| Method | BikeNYC | |
|---|---|---|
| | RMSE | MAE |
| ARIMA | 10.07 | 6.41 |
| SARIMA | 10.56 | 5.44 |
| VAR | 9.92 | 6.33 |
| DeepST | 7.43 | 4.25 |
| ST-ResNet | 6.33 | 4.03 |
| AttoConvLSTM | 7.09 | 4.19 |
| DMVST-Net | 6.01 | 3.95 |
| DST-ICRL | 5.93 | 3.11 |
| ACFM | 5.64 | - |
| ASTIR | 4.18 | - |
| SCDFNN(WM) | 3.78 | 2.23 |
| SCDFNN (modified WM) | **3.44** | **1.91** |

### 6.2.2. Ablation Experiment (Q2)

In order to evaluate the significance of each component of the model, we conducted ablation experiments, and present the resulting comparisons in Table 6. Specifically, we replaced the removed data with 0 values to ensure that the model's overall structure remained unaffected.

**Table 6.** Ablation experiment results.

| SCDFNN (Modified WM) | TaxiBJ | | BikeNYC | |
|---|---|---|---|---|
| | **RMSE** | **MAE** | **RMSE** | **MAE** |
| No closeness | 20.62 | 11.94 | 4.10 | 2.95 |
| No period | 17.12 | 10.69 | 3.45 | 1.92 |
| No trend | 17.46 | 10.74 | 3.44 | 1.91 |
| No Spatial | 17.79 | 10.80 | 3.50 | 1.94 |
| No external | 20.53 | 12.21 | 3.90 | 2.21 |
| Complete | 16.82 | 10.23 | 3.44 | 1.91 |

Comparing the complete SCDFNN with the "no external" and "no Spatial" models, we observe that external and spatial data contribute to the performance gains of the full model. Furthermore, the "no closeness" and "no external" models significantly underperform the complete SCDFNN, suggesting that the model extracts the most information from adjacent and external data.

Meanwhile, the "no period," "no trend," and "no spatial" models demonstrate slightly worse performance than the complete SCDFNN. This could be attributed to either: (1) less effective information being contained within these data; or (2) SCDFNN's inability to effectively extract information from these datasets. Given other deep neural network models' emphasis on spatial information, the latter is more probable.

6.2.3. Anti-Noise Experiment (Q3)

In order to assess the noise resistance of our model, we conducted experiments using Gaussian and Poisson perturbation on the TaxiBJ test set at varying noise intensities. The experimental results, shown in Figure 11, demonstrate the model's robustness to noise. Notably, we only added noise to the data of the spatial-temporal module and periodic module, while the external factors were kept accurate.

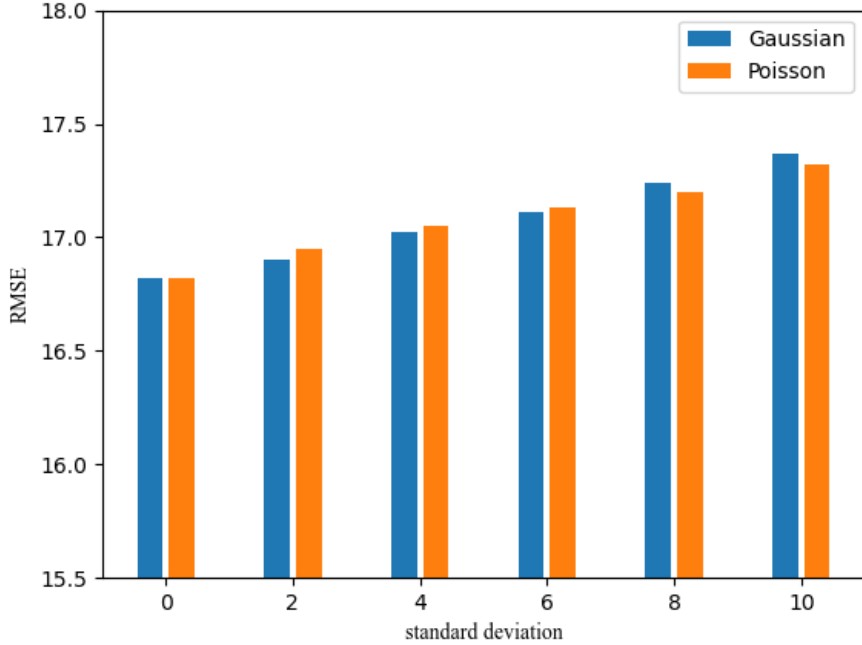

**Figure 11.** RMSE of TaxiBJ dataset under different intensities of Gaussian perturbation.

The *x*-axis of Figure 11 represents the standard deviation of the Gaussian and Poisson perturbations, with larger values indicating stronger noise interference, and 0 representing no noise. We observed that the model's accuracy remained stable under various intensities of noise. Even when adding Gaussian perturbation or Poisson perturbation with a standard

deviation of 10, the model's RMSE only increased by 3.27% or 2.97%. These results confirm the model's excellent anti-noise ability, which is due to the combination of multiple modules and fuzzy rules.

### 6.2.4. Training Time Comparison (Q3)

A comparison of the training time and root mean square error (RMSE) for the two Wang–Mendel methods, learned under different initial affiliation functions of the spatial-temporal module, is shown in Figure 12. It is apparent that the training time for the improved Wang–Mendel method is significantly reduced while maintaining accuracy comparable to that of the original Wang–Mendel method.

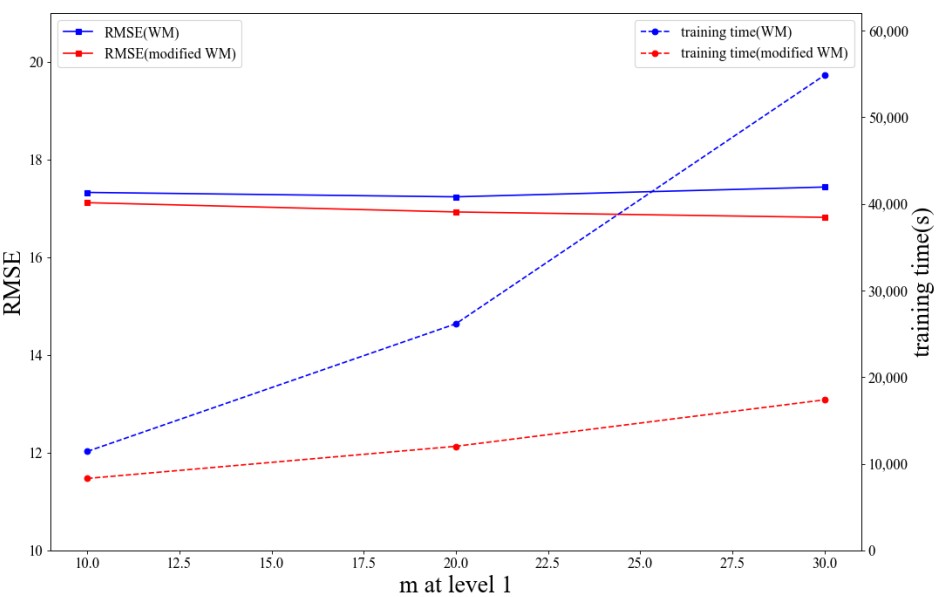

**Figure 12.** RMSE and training time of the model given different m.

As the value of m increases, the RMSE obtained by both methods is similar, with minimal change. However, the training time of the original Wang–Mendel method increases significantly. Specifically, when $m = 10$, the training time of the ordinary Wang–Mendel method is only 1.375 times that of the modified Wang–Mendel distribution. As m increases to 20 and 30, the ratio of training time further increases to 2.175 and 3.154, respectively. This finding demonstrates that the modified Wang–Mendel method effectively combines the membership function and prunes the fuzzy rules, resulting in a significant reduction in training time.

### 6.3. Interpretability Analysis (Q3)

Compared with ordinary neural networks, the biggest advantage of our model is its high interpretability. This interpretability is mainly reflected in two aspects: on the one hand, the rule parameters $c^{j_1,j_2,\ldots,j_n}$ of the model have clear mathematical meanings, and on the other hand, the model can easily track and correct erroneous predictions. Below, we will elaborate on the interpretability of SCDFNN from these two aspects, respectively.

Firstly, according to Equations (2), (11) and (12), it can be found that the rule parameter $c^{j_1,j_2,\ldots,j_n}$ of rule $(j_1, j_2, \ldots, j_n)$ in the model is designed as the weighted average of the output y of the samples falling on this rule, and the weight is equal to the activation strength $f$. A sample falling on a rule refers to the rule whose corresponding activation strength is the highest among all the rules for the sample. If a rule is not triggered by any sample, the extrapolation scheme in step seven of Section 3.2 is used to determine the rule parameter $c^{j_1,j_2,\ldots,j_n}$. Therefore, $c^{j_1,j_2,\ldots,j_n}$ can be regarded as an estimate of the expected output y of the fuzzy IF-THEN rule represented by $(j_1, j_2, \ldots, j_n)$. This reveals the clear

mathematical meaning of the rule parameters $c^{j_1, j_2, \dots, j_n}$, which is beneficial for researchers in better understanding the underlying principles of the model.

In addition, the interpretability of SCDFNN is also reflected in its ability to easily track and correct erroneous predictions, which is a capacity lacking in "black box" models. Specifically, for any input sample, each FNN will activate several rules, and we can easily identify the most influential rules among them, as shown in Figure 13. We can also represent them in IF-THEN form to help humans understand the model, as shown in Table 7. For the rules activated by erroneous samples, we can use the corresponding correct output data to update all rule parameters c according to the following equation to correct the model:

$$c_{new}^{j_1^*, j_2^*, \dots, j_n^*} = \alpha f^{j_1^*, j_2^*, \dots, j_n^*} y_{right} + \left(1 - \alpha f^{j_1^*, j_2^*, \dots, j_n^*}\right) c_{old}^{j_1^*, j_2^*, \dots, j_n^*}, \qquad (17)$$

where $(j_1^*, j_2^*, \dots, j_n^*)$ is the most influential rule triggered by the sample, $c_{old}^{j_1^*, j_2^*, \dots, j_n^*}$ is the original rule parameter, $f^{j_1^*, j_2^*, \dots, j_n^*}$ is the activation strength of the sample on this rule, $y_{right}$ is the correct output, $\alpha$ is the update weight, $0 < \alpha < 1$, the larger the value of $\alpha$, the greater the updating force, and cnew is the updated rule parameter.

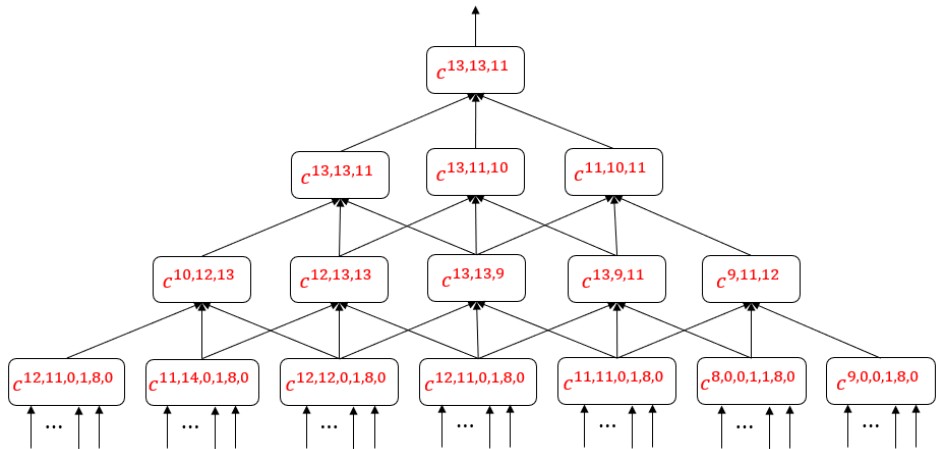

**Figure 13.** The rules triggered by a sample in the TaxiBJ dataset.

**Table 7.** IF-THEN rule triggered by an input of TaxiBJ dataset in the first layer of SCDFNN.

| Number | Meaning of Rule |
|:---:|:---:|
| 1 | IF $neighbor1_{6,5,\tau-2}^{out}$ is $A_1^{12}$ and $neighbor1_{6,5,\tau-1}^{out}$ is $A_2^{11}$ and today is not a holiday, and it′s a weekday and $time_\tau$ is $A_5^8$ and $Weather_{\tau-1}$ is $A_6^0$, THEN $y_1^1$ is $c^{12,11,0,1,8,0}$ |
| 2 | IF $neighbor2_{6,5,\tau-2}^{out}$ is $A_7^{11}$ and $neighbor2_{6,5,\tau-1}^{out}$ is $A_8^{14}$ and today is not a holiday, and it′s a weekday and $time_\tau$ is $A_{11}^8$ and $Weather_{\tau-1}$ is $A_{12}^0$, THEN $y_2^1$ is $c^{11,14,0,1,8,0}$ |
| 3 | IF $X_{6,5,\ \tau-4}^{in}$ is $A_{13}^{12}$ and $X_{6,5,\ \tau-3}^{in}$ is $A_{14}^{12}$ and today is not a holiday, and it′s a weekday and $time_\tau$ is $A_{17}^8$ and $Weather_{\tau-1}$ is $A_{18}^0$, THEN $y_3^1$ is $c^{12,12,0,1,8,0}$ |
| 4 | IF $X_{6,5,\ \tau-3}^{in}$ is $A_{19}^{12}$ and $X_{6,5,\ \tau-2}^{in}$ is $A_{20}^{11}$ and today is not a holiday, and it′s a weekday and $time_\tau$ is $A_{23}^8$ and $Weather_{\tau-1}$ is $A_{24}^0$, THEN $y_4^1$ is $c^{12,11,0,1,8,0}$ |
| 5 | IF $X_{6,5,\ \tau-2}^{in}$ is $A_{25}^{11}$ and $X_{6,5,\ \tau-1}^{in}$ is $A_{26}^{11}$ and today is not a holiday, and it′s a weekday and $time_\tau$ is $A_{29}^8$ and $Weather_{\tau-1}$ is $A_{30}^0$, THEN $y_5^1$ is $c^{11,11,0,1,8,0}$ |
| 6 | IF $X_{6,5,\ \tau-48}^{in}$ is $A_{31}^8$ and one day ago is not a holiday, and it′s a weekday and today is not a holiday, and it′s a weekday and $time_\tau$ is $A_{36}^8$ and $Weather_{\tau-1}$ is $A_{37}^0$, THEN $y_6^1$ is $c^{8,0,0,1,1,8,0}$ |
| 7 | IF $X_{6,5,\ \tau-48*7}^{in}$ is $A_{38}^9$ and a week ago was not a holiday and today is not a holiday, and it′s a weekday and $time_\tau$ is $A_{42}^8$ and $Weather_{\tau-1}$ is $A_{43}^0$, THEN $y_7^1$ is $c^{9,0,0,1,8,0}$ |

## 7. Conclusions

We have proposed a self-built deep fuzzy neural network model, named SCDFNN and based on fuzzy theory, to address the lack of interpretability in models in the intelligent transportation field. Our model is more interpretable than are traditional deep neural network models, which provides clearer and more intuitive prediction results for intelligent transportation systems. This can better assist the government with congestion prediction, accident prediction and road network planning. Additionally, the interpretable SCDFNN model can help the public to better understand and accept the decisions made by policymakers and government departments. We evaluated our proposed model using two publicly-available and widely-used datasets, and the results showed that our model outperformed other existing methods on the BikeNYC dataset.

We also improved the Wang–Mendel method by incorporating pruning ideas and utilized the modified Wang–Mendel method to train our model. The experiment demonstrated that our modified method's training process was over twice as fast as the original, while maintaining accuracy. Additionally, we performed anti-noise experiments, and the results revealed that our model exhibited excellent anti-noise ability. Finally, we have explained the model's mathematical significance and its capability to track precise errors.

However, our SCDFNN model exhibited weak spatial information extraction capabilities, which hindered its performance on the TaxiBJ dataset. Additionally, the model only used triangular membership functions, which limited its potential. Future research might explore combining the spatial information extraction capabilities of the GCN model with the SCDFNN model, or exploring different membership function forms.

**Author Contributions:** Conceptualization, J.A. and J.Z.; methodology, J.A. and J.Z.; software, J.Z. and X.Q.; validation, J.Z. and X.Q.; formal analysis, J.Z. and Q.L.; investigation, J.Z. and J.C.; resources, J.Z.; data curation, J.Z.; writing—original draft preparation, J.Z., Q.L. and X.Q.; writing—review and editing, J.Z., Q.L. and X.Q.; visualization, J.Z.; supervision, J.A.; project administration, J.A. and J.Z.; funding acquisition, J.C. All authors have read and agreed to the published version of the manuscript.

**Funding:** This study was supported in part by the National Natural Science Foundation of China under Grant 62172147, and the Natural Science Foundation of Hunan Province, China under Grant 2018JJ2063, and it was also supported in part by the innovation project of Hunan Xiangjiang Artificial Intelligence Academy.

**Data Availability Statement:** The TaxiBJ and BikeNYC datasets can be obtained at https://github.com/aptx1231/NYC-Dataset (accessed on 23 February 2023).

**Conflicts of Interest:** The authors declare no conflict of interest.

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
