# Peer review of "Self-Constructed Deep Fuzzy Neural Network for Traffic Flow Prediction"

_electronics, doi:10.3390/electronics12081885_

Round 1

Reviewer 1 Report

Comments and Suggestions for Authors

Intelligent transportation systems estimate traffic flow to reduce congestion. In the deep learning-based traffic cognition age, traffic forecast interpretability must be improved for persuasiveness. Several research projects have tackled interpretable neural networks from the feature and outcome levels. Still, the model-level explanation, which reveals traffic prediction theory through transparent models, is underexplored and deserves more attention. Traffic flow prediction and model interpretability are proposed using a self-constructed deep fuzzy neural network (SCDFNN). Due to recent advances in neuro-symbolic computation for automatic rule learning, the SCDFNN learns interpretable human traffic cognitive rules (consisting of driver and traffic attributes) based on a deep understanding with two innovations: (1) a novel fuzzy neural network hierarchical architecture for spatiotemporal dependences is constructed in the traffic feature domain; (2) a modified Wang-Mendel method is used to fuse with regional differences in traffic da. The SCDFNN is transparent and comparable to state-of-the-art deep models in extensive experiments on well-known traffic datasets. In conclusion, the reviewer found that the paper has merits and could be acceptable to publish in future forms.

Therefore, please revise the manuscript according to the reviewer's comments. - Generally speaking, the contents of this manuscript meet all requirements for scientific papers.

The authors offered a novel technique with numerous favorable outcomes, and the article has a scientific tone and style. In addition to this study, the authors' future studies might consider conducting empirical research.

Here are the reviewer's comments

- Please add a blank between "Fig." and "the figure number." Please see lines 239, 297, 333, 362, 374, 383, 403, 409, 419, 480, 585, 595, and 632. - Line 534: "Tables III and IV" should be Tables 3-4.

- Line 565: Table V should be Table 5.

- Line 632: TABLE VI should be Table 6.

- All equations should be mentioned or explained in the text.

- Please provide the flowchart for the authors' proposed method.

- Please add the application of the machine learning algorithms (Fuzzy Neutral Networks) for different fields, which should introduce in your literature review/related works, and limit this study to a specific task. Algorithms are currently used in many areas: Please consider adding the following good fit references for the design concepts which are mentioned in your literature review, "Optimum Design for the Magnification Mechanisms Employing Fuzzy Logic– ANFIS"; " Efficient road traffic anti-collision warning system based on fuzzy nonlinear programming" and " Optimum Design for the Magnification Mechanisms Employing Fuzzy Logic–ANFIS" for more references that could be used to enrich your literature review.

- Citations in the text and references should be followed the journal template. The reviewer suggests you search the Journal of Marine Science and Engineering or other Journals. Generally, a manuscript (average) will have about 40-50 papers. References should be numbered in order of appearance and indicated by a numeral or numerals in square brackets—e.g., [1] or [2,3], or [4–6]. In the text, reference numbers should be in square brackets [ ] and placed before the punctuation; for example, [1], [1–3], or [1,3]. For embedded citations in the text with pagination, use both parentheses and brackets to indicate the reference number and page numbers; for example, [5] (p. 10) or [6] (pp. 101–105).

- The applications of machine learning algorithms to predict the traffic flow will increase the calculation costs, complicate the situation, and increase the calculation time. How did the authors balance the two economic and technical issues in the trade-off of the proposed method?

- How do the authors validate the results?

- What is the main limitation of this study, and what is your further research? Please explain in Section 7.

The reviewer hopes his point of view could help the authors improve their work well and highly appreciates your work. 

Thank you for reading.

Author Response

Comments: Please add a blank between "Fig." and "the figure number." Please see lines 239, 297, 333, 362, 374, 383, 403, 409, 419, 480, 585, 595, and 632. Line 534: "Tables III and IV" should be Tables 3-4. Line 565: Table V should be Table 5. Line 632: TABLE VI should be Table 6.

Response: Sorry for this tiny negligence, we have revised the citation format of the figures and tables in the paper.

Comments: All equations should be mentioned or explained in the text.

Response: We have supplemented the explanation of all the equations.

Comments: Please provide the flowchart for the authors' proposed method.

Response: We provide a flowchart for the modified Wang-Mendel method. Please see Fig. 3 on page 9 of the revised document.

Comments: Please add the application of the machine learning algorithms (Fuzzy Neutral Networks) for different fields, which should introduce in your literature review/related works, and limit this study to a specific task. Algorithms are currently used in many areas: Please consider adding the following good fit references for the design concepts which are mentioned in your literature review, "Optimum Design for the Magnification Mechanisms Employing Fuzzy Logic– ANFIS"; " Efficient road traffic anti-collision warning system based on fuzzy nonlinear programming" and " Optimum Design for the Magnification Mechanisms Employing Fuzzy Logic–ANFIS" for more references that could be used to enrich your literature review.

Response: We have supplemented the application of FNN in different fields in related works of the revised paper, and have limited this study to a specific task. Additionally, we have carefully read the articles you recommended and cited them in the revised paper.

Comments: Citations in the text and references should be followed the journal template. The reviewer suggests you search the Journal of Marine Science and Engineering or other Journals. Generally, a manuscript (average) will have about 40-50 papers. References should be numbered in order of appearance and indicated by a numeral or numerals in square brackets—e.g., [1] or [2,3], or [4–6]. In the text, reference numbers should be in square brackets [ ] and placed before the punctuation; for example, [1], [1–3], or [1,3]. For embedded citations in the text with pagination, use both parentheses and brackets to indicate the reference number and page numbers; for example, [5] (p. 10) or [6] (pp. 101–105).

Response: Sorry for this tiny negligence, we have now modified the citation format in our paper according to the journal template.

Comments: The applications of machine learning algorithms to predict the traffic flow will increase the calculation costs, complicate the situation, and increase the calculation time. How did the authors balance the two economic and technical issues in the trade-off of the proposed method?

Response: The SCDFNN model we propose uses a modified Wang-Mendel method for training, which is a non-iterative algorithm that only requires one pass through the training data. Therefore, compared to machine learning models that use traditional iterative learning algorithms such as gradient descent, the training cost of the SCDFNN model is very low. Moreover, the SCDFNN predicts traffic flow over a longer time span (30 minutes to 1 hour), and the model's computation time is much shorter compared to the prediction time span, so its calculation cost can be ignored.

Comments: How do the authors validate the results?

Response: We split the TaxiBJ and BikeNYC datasets into training and testing sets and used supervised learning on the training data to train the model while controlling variables to select the optimal parameters. We used RMSE and MAE as metrics and compared our model with 12 baseline models. The results showed that our model outperformed all other models on the BikeNYC dataset. In addition, we conducted ablation experiments, noise resistance experiments, and comparison experiments between the modified Wang-Mendel method and the original method to validate the various performance aspects of the model. Please refer to section 6.2 for specific results.

Comments: What is the main limitation of this study, and what is your further research? Please explain in Section 7.

Response: We have rewritten the conclusion section to address these questions. Please see Section 7 on pages 21-22 of the revised paper.

Reviewer 2 Report

 In this paper, authors have proposed a new type of self-constructed deep fuzzy neural network (SCDFNN) for traffic flow prediction. The work proposed is interesting, however, there are few major concerns that must be addressed before it can be considered for publishing. I offer the following comments to the authors.

C#1: Consider reducing the abstract and add specific findings of the current work.

C#2: Highlight the importance/value of traffic flow prediction and its role in mitigating the congestion problem in urban areas. It is recommended to further enrich the introduction and related works section by reviewing and citing the following important and latest studies: doi.org/10.3390/s20030685; DOI: 10.1109/BIGCOMP.2017.7881687; doi.org/10.3390/su12020646; doi.org/10.1080/23311916.2021.2010510 doi.org/10.3390/s22186921;

C#3: In lines, 112-113 authors mentioned that “To the best of our knowledge, this is the first work utilizing deep fuzzy neural net-112 works for traffic flow prediction”, however, this is not true as several recent works have adopted application of different DNs for traffic flow prediction. Authors should be  careful in using such statements.

C#4: Forecasting traffic flow for relatively longer time spans such as 30 minutes to one hour will not aid any significant value to resolve dynamic traffic congestion issue, give your opinion about this?

C#5: In section 3.1. it is suggested to review and cite the following paper for the structure and working mechanism of FNN. https://doi.org/10.1016/j.iatssr.2022.08.003

C#6: there are several symbols and formulae used in the methods section as well as throughout the paper, it is recommended to define all the variables clearly. In this regard a table of symbols may be included.

C#7: The paper mentions the use of spatio-temporal architecture, but Figure 6 seems to depict only the prediction of temporal flow. Can you explain why the authors did not add a location-based prediction figure? Further, visualization inflow and outflow for February 2016 is provided, why not from the latest years?

C#8:  Why did author select 30 minutes and 1 hour time intervals for the analysis? Further, what is the rationale behind comparing two different modes from two different contexts (cities) i.e.,  TaxiBJ and BIKENYC? Drivers characteristics, roadway conditions, and weather patterns vary from one location to other, so the provided comparison does not make much sense. This point should be specifically addressed with more clarity.

C#9:  It is suggested that authors should rewrite the conclusion part. It may commence with explaining the purpose and what has been done previously in the domain of studies. Also, provide future research directions precisely in the respective domain and suggest relevant practical recommendations for practitioners the society and research domains.

C#10: table 1 dataset used contains only few variables that can influence traffic flow prediction, in the field the flow is affected by several other parameters and neglecting them may include an element of bias in the obtained results.

C#11: Section 6.3, the interpretability analysis explanation is vague and unclear and need a thorough an overhaul to make it easier to understand and interpret.

C#12: the paper has several typos and language issues and read a careful language audit by a native speaker.  

Author Response

Comments: Consider reducing the abstract and add specific findings of the current work.

Response: We have simplified the abstract and emphasized the specific findings of the study to summarize the research contribution more clearly, while avoiding excessive background and detail information.

Comments: Highlight the importance/value of traffic flow prediction and its role in mitigating the congestion problem in urban areas. It is recommended to further enrich the introduction and related works section by reviewing and citing the following important and latest studies: doi.org/10.3390/s20030685; DOI: 10.1109/BIGCOMP.2017.7881687; doi.org/10.3390/su12020646; doi.org/10.1080/23311916.2021.2010510 doi.org/10.3390/s22186921;

Response: Thank you very much for your suggestion. We emphasized the importance and value of traffic flow prediction, as well as its critical role in mitigating the congestion problem in urban areas. Please see the revised Introduction. Additionally, we have carefully read the articles you recommended and cited them in the revised paper.

Comments: In lines, 112-113 authors mentioned that “To the best of our knowledge, this is the first work utilizing deep fuzzy neural net-112 works for traffic flow prediction”, however, this is not true as several recent works have adopted application of different DNs for traffic flow prediction. Authors should be  careful in using such statements.

Response: Thank you very much for your reminder. We have confirmed this and deleted the relevant parts in the revised paper. We deeply apologize for these omissions.

Comments: Forecasting traffic flow for relatively longer time spans such as 30 minutes to one hour will not aid any significant value to resolve dynamic traffic congestion issue, give your opinion about this?

Response: We agree with the view that predicting traffic flow over a relatively longer time span may not be the most effective way to solve the problem of dynamic traffic congestion, as traffic patterns can change quickly and unpredictably. However, we believe that it can still be useful in certain situations, such as traffic planning, infrastructure development, and policy making, to reduce congestion in the long term.

Comments: In section 3.1. it is suggested to review and cite the following paper for the structure and working mechanism of FNN. https://doi.org/10.1016/j.iatssr.2022.08.003

Response: We have supplemented a review of the structure and working mechanism of FNN in Section 3.1 of the revised paper. In addition, we have carefully read and cited the article you recommended.

Comments: there are several symbols and formulae used in the methods section as well as throughout the paper, it is recommended to define all the variables clearly. In this regard a table of symbols may be included.

Response: We have clarified the meaning of all symbols and added a symbol table. Please see to Table 1 on page 6 of the revised paper.

Comments: The paper mentions the use of spatio-temporal architecture, but Figure 6 seems to depict only the prediction of temporal flow. Can you explain why the authors did not add a location-based prediction figure? Further, visualization inflow and outflow for February 2016 is provided, why not from the latest years?

Response: We supplemented the flow heatmap of TaxiBJ dataset on February 13, 2016 to observe the flow characteristics of different locations, as shown in Figure 7(b) on page 12. In this study, we divided the flow prediction problem into multiple single-output prediction problems according to the size of the grid map, as FNN is suitable for single-output problems. Each prediction problem only extracts spatial dependency information from adjacent areas, so the focus of the model is on time series prediction rather than location-based prediction, as detailed in sections 3.3 and 4.1. In addition, the two datasets used in the experiment were released with the publication of Junbo Zhang et al.'s article "Deep Spatio-Temporal Residual Networks for Citywide Crowd Flows Prediction" and have been widely used by related researchers since then. To better compare with their proposed model, we chose these two datasets.

Comments: Why did author select 30 minutes and 1 hour time intervals for the analysis? Further, what is the rationale behind comparing two different modes from two different contexts (cities) i.e., TaxiBJ and BIKENYC? Drivers characteristics, roadway conditions, and weather patterns vary from one location to other, so the provided comparison does not make much sense. This point should be specifically addressed with more clarity.

Response: The main reason we choose 30-minute and 1-hour time intervals for analysis is to balance model complexity, interpretability, and prediction accuracy. The shorter time interval data has higher randomness and complexity, which makes it difficult for the model to maintain both interpretability and high accuracy. On the other hand, longer time intervals reduce the number of samples and lower the research value. Additionally, comparing data from different cities and modes can help us understand the generality and applicability of traffic flow prediction, and also help us gain insight into the characteristics and patterns of different traffic modes. Furthermore, this can also improve the stability and generalization ability of the model, avoiding the limitations of prediction models due to the specificity of data.

Comments: It is suggested that authors should rewrite the conclusion part. It may commence with explaining the purpose and what has been done previously in the domain of studies. Also, provide future research directions precisely in the respective domain and suggest relevant practical recommendations for practitioners the society and research domains.

Response: Thank you very much for your suggestion. We have rewritten the conclusion section to supplement future research directions. Please refer to section 7 on pages 21-22 of the revised paper.

Comments: table 1 dataset used contains only few variables that can influence traffic flow prediction, in the field the flow is affected by several other parameters and neglecting them may include an element of bias in the obtained results.

Response: We have added some variables to the table that may affect traffic flow prediction to reduce the bias in the results. Please see Table 2 on page 15 of the revised paper. It should be noted that both datasets were released with the publication of the article "Deep Spatio-Temporal Residual Networks for Citywide Crowd Flows Prediction" by Junbo Zhang et al., and have been widely used in subsequent related studies. We do not have any additional data details.

Comments: Section 6.3, the interpretability analysis explanation is vague and unclear and need a thorough an overhaul to make it easier to understand and interpret.

Response: We have rewritten the analysis of model interpretability in Section 6.3 to make it easier to understand.

Comments: the paper has several typos and language issues and read a careful language audit by a native speaker.

Response: We have thoroughly polished the linguistic part to improve the readability of this article, in which we revised also the whole manuscript carefully to avoid grammatical mistakes and unusual expressions.

Reviewer 3 Report

Recommendation: Major Revision

The topic discusses the self-constructed deep fuzzy neural network for traffic flow prediction. This article discusses the importance of traffic flow prediction in preventing traffic congestion, especially in urban road environments. The authors have developed a deep fuzzy neural network that not only provides high accuracy, but also better interpretability for traffic prediction.

 1.        What is the importance of interpretability in traffic prediction?

 2.        How does the deep fuzzy neural network address this problem?

 3.        Can you explain how the authors trained and tested their neural network and what the results were?

 4.    How can the results of this research be applied to real traffic management systems?

5. The conclusion provides a clear summary of the proposed model, its performance, and the modifications made to the Wang-Mendel algorithm. However, it would be useful to provide more detail on the implications of the proposed model for practical applications in intelligent transport systems and how it could be integrated into existing traffic management systems.

Author Response

Comments: What is the importance of interpretability in traffic prediction?

Response: In traffic prediction, interpretability plays three important roles. Firstly, interpretable models can provide clearer and more intuitive prediction results, which can better assist the government in subsequent applications such as congestion prediction, accident prediction, and road network planning. Secondly, interpretable models can help the public better understand and accept the decisions of the management department, and are more likely to pass government and ethical reviews. Finally, interpretability provides reasons for how the model arrives at prediction results, which can help researchers understand the model principles and further derive technological developments.

Comments: How does the deep fuzzy neural network address this problem?

Response: DFNN is a combination of fuzzy system and neural network, which integrates the advantages of both. DFNN can optimize model parameters using data-driven hybrid algorithms, and its model structure enables it to be interpreted as a set of fuzzy rules that are easy for humans to understand, providing insights into the model decision-making process. Therefore, DFNN has high interpretability. In Section 3.1, we explain how FNN embodies fuzzy rules.

Comments: Can you explain how the authors trained and tested their neural network and what the results were?

Response: We split the TaxiBJ and BikeNYC datasets into training and testing sets and used supervised learning on the training data to train the model while controlling variables to select the optimal parameters. We used RMSE and MAE as metrics and compared our model with 12 baseline models. The results showed that our model outperformed all other models on the BikeNYC dataset. In addition, we conducted ablation experiments, noise resistance experiments, and comparison experiments between the modified Wang-Mendel method and the original method to validate the various performance aspects of the model. Please refer to section 6.2 for specific results.

Comments: How can the results of this research be applied to real traffic management systems?

Response: The interpretable SCDFNN model can be widely applied in practical traffic management systems, such as providing clearer and more intuitive prediction results for intelligent traffic systems, thus better assisting governments in congestion prediction, accident prediction, and road network planning. In addition, the interpretable SCDFNN model can help the public better understand and accept the decisions made by management departments. Overall, integrating this model into existing traffic management systems can make traffic management more efficient and intelligent.

Comments: The conclusion provides a clear summary of the proposed model, its performance, and the modifications made to the Wang-Mendel algorithm. However, it would be useful to provide more detail on the implications of the proposed model for practical applications in intelligent transport systems and how it could be integrated into existing traffic management systems.

Response: Thank you very much for your suggestions. We have rewritten the conclusion section and supplemented the impact of the SCDFNN model on the practical application of intelligent transportation systems, as well as answered how to integrate the SCDFNN model into existing traffic management systems. Please see section 7 on pages 21-22 of the revised document.

Round 2

Reviewer 2 Report

Authors have addressed my previous comments well. I believe the paper is now ready for acceptance in the journal. 

Reviewer 3 Report

Recommendation: Accept